# Female mice ultrasonically interact with males during courtship displays

**Joshua P Neunuebel[1,2]\*, Adam L Taylor[1], Ben J Arthur[1], SE Roian Egnor[1]\***

[1]Janelia Research Campus, Howard Hughes Medical Institute, Ashburn, United States; [2]Department of Psychological and Brain Sciences, University of Delaware, Newark, United States

**Abstract** During courtship males attract females with elaborate behaviors. In mice, these displays include ultrasonic vocalizations. Ultrasonic courtship vocalizations were previously attributed to the courting male, despite evidence that both sexes produce virtually indistinguishable vocalizations. Because of this similarity, and the difficulty of assigning vocalizations to individuals, the vocal contribution of each individual during courtship is unknown. To address this question, we developed a microphone array system to localize vocalizations from socially interacting, individual adult mice. With this system, we show that female mice vocally interact with males during courtship. Males and females jointly increased their vocalization rates during chases. Furthermore, a female's participation in these vocal interactions may function as a signal that indicates a state of increased receptivity. Our results reveal a novel form of vocal communication during mouse courtship, and lay the groundwork for a mechanistic dissection of communication during social behavior.

## Introduction

Vocalizations are a crucial part of courtship and mating in a wide variety of species (*Bradbury and Vehrencamp, 1998*). Gazelles roar during mating displays (*Blank et al., 2014*), songbirds stimulate reproductive activity in females by singing (*Bentley et al., 2000*), sac-winged bats produce complex serenades during sexual displays (*Behr and von Helversen, 2004*), and anurans phonotaxis towards ultrasonic courtship vocalizations (*Shen et al., 2008*). In mice (*Mus musculus*), several indirect observations suggest that ultrasonic vocalizations play a role in courtship. For example, USVs occur during mating (*Sewell, 1972*; *Hanson and Hurley, 2012*), urine from females but not males stimulates vocal production in males (*Nyby et al., 1979*; *Musolf et al., 2010*), female mice spend more time near vocalizing males than mute males (*Pomerantz et al., 1983*), and female mice preferentially approach a speaker playing male vocalizations rather than a speaker playing background noise (*Musolf et al., 2010*) or whistle-like control sounds (*Hammerschmidt et al., 2009*).

When examining the ultrasonic vocal behavior of adult mice, most experimental paradigms use pairs of mice (*Bean et al., 1981*; *Choi et al., 2011*; *D'Amato and Moles, 2001*; *Hammerschmidt et al., 2012b*; *Nunez et al., 1978*; *Nyby et al., 1976*; *Ogawa et al., 2000*; *Pomerantz and Clemens, 1981*; *Rotschafer et al., 2012*; *Scattoni et al., 2008*; *Warburton et al., 1989*; *White et al., 1998*). However, mice are adaptable and opportunistic animals that thrive in diverse conditions. Mice can occupy large and sparsely populated territories, where dyadic interactions would be most common, or small, densely inhabited regions where multiple individuals would frequently interact (*Anderson, 1961*; *Berry and Bronson, 1992*). *Portfors (2007)* reported that USV rate and complexity increase when mice are housed in large, mixed sex groups. Moreover, mice living in large populations for extended periods exhibit social interactions that are affected by the dynamics of multiple group members (*Shemesh et al., 2013*) and complex dominance hierarchies (*Weissbrod et al., 2013*). These interactions may have reproductive consequences that impact courtship by forcing females to choose

\*For correspondence:
jneunuebel@psych.udel.edu
(JPN); egnorr@janelia.hhmi.org
(SERE)

Competing interests: The authors declare that no competing interests exist.

**eLife digest** Male songbirds are famous for using singing to attract mates, and many other animals also make noises as part of their mating rituals. Male mice, for example, are known to make high-pitched noises as they court females—but these noises are beyond the range of human hearing.

The scent of female mice is enough to cause male mice to start 'singing', and female mice are attracted to the male's song. Many studies have examined how these songs affect the mating behavior of the mice. But it was not always easy for a researcher to tell which mouse in the pair was making the sounds, because the mice make no obvious movements when they sing.

Neunuebel et al. have now used many microphones arranged closely together to accurately pinpoint the singing individuals in a cage full of mice. Unexpectedly, this approach revealed that female mice join the males in song when courting. This challenges the long held belief that only male mice sing to communicate their courtship interests, and suggests that previous studies may have incorrectly attributed sounds from female mice as coming from males instead.

Neunuebel et al. showed that as a male mouse pursues a female, both may sing. When females sing, they slow down; this makes it easier for the male to catch them. Females who don't respond to the male's song, however, keep up the pace. This suggests the song may be a way for the female mouse to indicate that she is interested in mating with her suitor.

The next step will be to measure whether the noises that a female makes change depending on the identity of the suitor. Previous research has shown that female mice have more and healthier pups when they are paired with a male that they choose as opposed to a random male. The same microphone array could now be used to see whether female mice 'say' different things when paired with a male they like versus a male they don't.

between males (*Silk, 2007*). Therefore, examining the vocal repertoire of groups of mice housed together is a unique, untapped opportunity to examine USV production during complex social contexts. These conditions may provide insight into the role vocalizations play in mouse social behavior.

Identifying the source of a mouse ultrasonic vocalization during social interactions between multiple potential vocalizers is challenging. When mice produce USVs, there are no visible movements associated with these signals (*Chabout et al., 2012*) and the vocalizations are inaudible to humans. Males are believed to produce the majority of vocalizations during male-female encounters. Previous reports have shown that female mice rarely vocalize in the presence of an anesthetized or devocalized male, and more often than not are completely silent (*Whitney et al., 1973*; *Warburton et al., 1989*). However, females have been reported to vocalize at high rates during same sex interactions (*Maggio and Whitney, 1985*; *Moles et al., 2007*). Therefore, females may be vocalizing during encounters with males when animals are intact and freely behaving. To further complicate our ability to identify the source of a vocalization, these signals are similar across individuals, both within and between sexes (*Hammerschmidt et al., 2012a*). These complications require a technological advance to unmask the vocal contribution of each individual mouse.

To investigate the role of ultrasonic vocalizations during social behavior, we developed a microphone-array-based system for localizing the source of USVs in socially interacting groups of mice. The system has three components: (1) estimating the likelihood that the vocalization was produced at any position in the cage, (2) automatically determining the spatial position of every mouse in the cage at the time of the vocalization, and (3) combining the positional information of the mice and sound source estimate to assign a vocalization to a specific mouse. First, we show that the system precisely and accurately assigns vocalization to mice. After determining the resolution of the system, we investigated the vocal behavior of groups of male and female mice during courtship. Using the array, we revealed that both sexes vocalize, contrary to the belief that courtship vocalizations are male-specific. Furthermore, we discovered a temporal correlation between male and female vocalizations. Vocal interactions were observed between all possible male-female pairs in the colony. Moreover, these vocal exchanges were predominantly detected during chases. Finally, females that vocally interacted with pursuing males had lower chase speeds than silent females, suggesting a role for these exchanges in communicating female receptivity.

## Results

### Development of a microphone array

A four-channel ultrasonic microphone array based system was developed to identify vocalizing mice using a procedure modified from *Zhang et al. (2008)*. Video and audio data were synchronously recorded during experiments (*Figure 1A–B*). Following the experiment, the video-based movement trajectory was determined for each subject (*Ohayon et al., 2013*). Vocal signals were automatically extracted (see 'Materials and methods'; *Figure 1B–C*). To estimate the location of a vocal signal, each signal was partitioned into time-frequency snippets—filtered pieces of the signal 5 ms long and 2 kHz wide (*Figure 1D*). For each snippet, a single location was found that best explained the different time delays observed between all possible microphone pairs (*Figure 1E–F*). Multiple estimates from the same vocal signal were then averaged to estimate the location of the sound source (*Figure 1G*).

Determining the vocal contribution of a specific mouse when recording from groups of animals requires a systematic way to assign the vocal signal to an individual. To achieve this, we developed a method that combined the positions of the mice at the time of the signal with the likelihood that the signal was emitted at any given position in the cage. Since every vocal signal used multiple estimates to determine the location of the sound source, we were able to calculate the probability density across the cage. Each mouse was assigned a density from the probability density function depending on the position of the mouse at the time of the vocal signal. Based on the assumption that a vocal signal was emitted from one of the recorded mice, a mouse probability index (MPI) was calculated for each mouse using the following formula:

$$MPI_n = \frac{D_n}{\sum_{i=1}^{M} D_i},$$

where $n$ = mouse index and $M$ = total number of mice. Consequently, we were able to calculate the likelihood that any animal in the arena emitted the vocal signal by combining information from the microphone array and the mouse position trajectories.

The accuracy and resolution of the system was evaluated by recording female-urine-elicited USVs from individual male mice. This approach ensured that all vocal signals were from an identified source and that the position of the source was known at the time of every vocal signal. In six recording sessions, we detected a total of 3724 vocal signals, of which 2590 were localizable. Localization was restricted to signals that contained at least three snippets, which provided the minimum amount of data to calculate the probability density function across the cage. We then created a simulated social environment by randomly generating the positions of three virtual mice at the time of each vocal signal. The locations of the virtual mice were confined to an area within the boundaries of the cage. Using this technique directly allowed us to quantify the accuracy and precision of the system under tightly controlled conditions.

The level of certainty associated with assigning a vocal signal to a mouse is related to the positions of the mice and the location of estimated sound source. Since the MPI describes the relative likelihood that a mouse emitted the vocal signal, the animal closest to the estimated sound source should in theory have the largest MPI. To examine this relationship, we compared each real and virtual animal's MPI with the error between the estimated sound source location and the animal. Error was defined as the distance between the estimated sound source location and the true source position. *Figure 2A* shows a negative correlation between MPI and error (r = −0.66; p < 10$^{-5}$), indicating that mice closer to the estimated sound source location have higher MPI values.

The accuracy of the system is directly related to the level of certainty associated with assigning a vocal signal to a mouse. To determine the effect of MPI magnitude on the proportion of localizable signals that were correctly assigned, we incorporated a threshold prior to assigning the vocal signal to either a real or virtual mouse. *Figure 2B* shows that increasing the MPI threshold increased the accuracy of the system, with a threshold of 0.95 causing the vocal signal to correctly be assigned to the real mouse 97.0% of the time. However, the MPI threshold was inversely related to the number of assigned vocalizations, indicating that the increase in accuracy came at the cost of a decrease in the number of assigned vocalizations (*Figure 2B*; see 'Materials and methods' for additional details).

The resolution of the system was evaluated on vocal signals that were assigned to individual mice after meeting our inclusion criteria. When determining the system's resolution, the median error

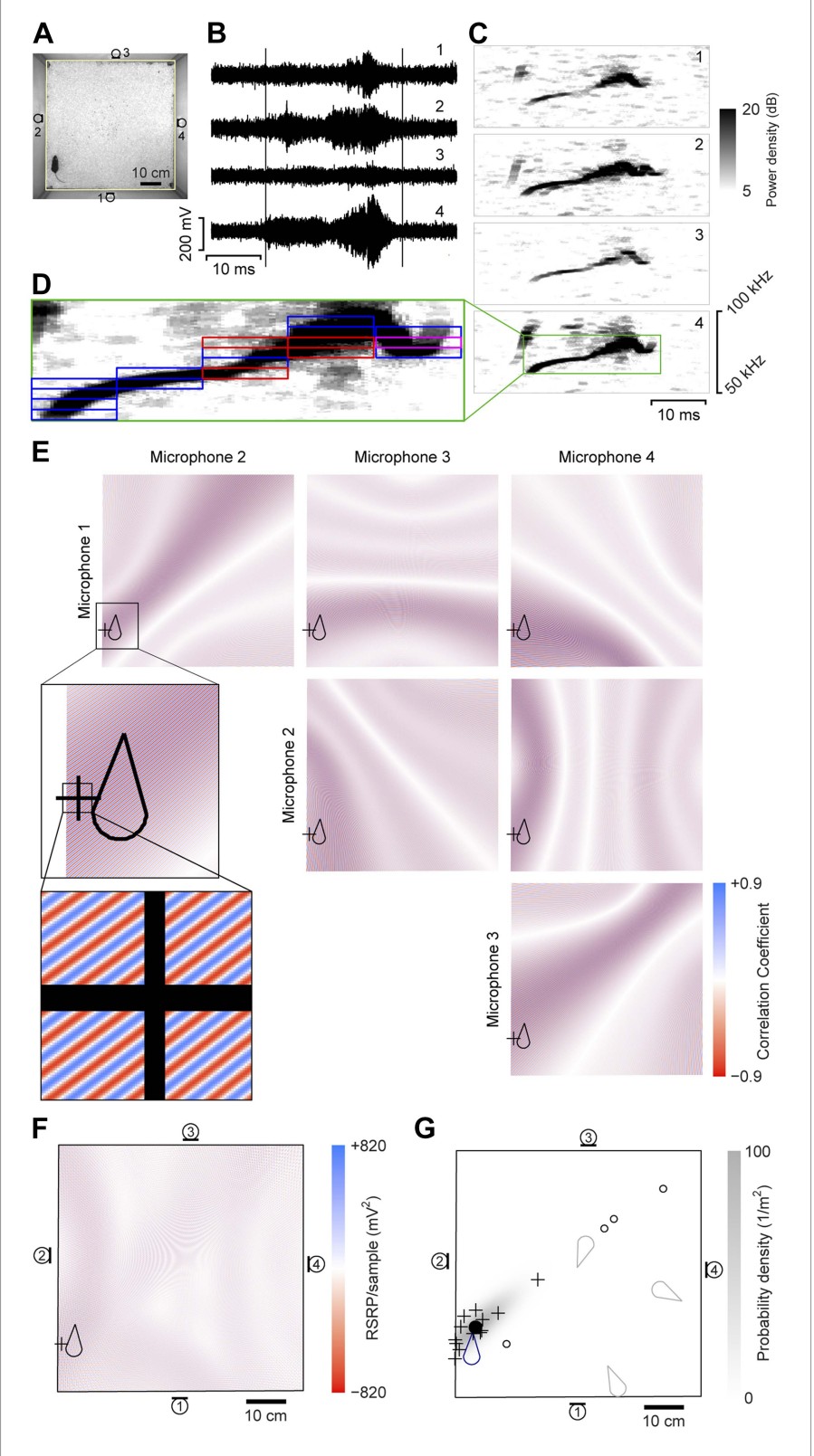

**Figure 1**. Illustration of sound-source localization procedure. (**A**) Image shows the location of a mouse in the behavioral arena during one video frame. Microphone locations are indicated by numbered microphone symbols (a circle with a tangent line segment). Yellow quadrilateral indicates the floor boundaries. (**B**) Vocal signal recorded at the same time as the frame in panel **A**. The number of each signal corresponds to the microphone in **A**.

*Figure 1. continued on next page*

*Figure 1. Continued*

Vertical lines indicate the start and end of the signal extracted by the audio segmentation software. (C) Spectrograms of the signals in **B**. Numbers in upper-right corner indicate the corresponding microphone. In the fourth microphone spectrogram, the large green rectangle indicates the time- and frequency-bounding box determined by the audio segmentation software. (D) Smaller rectangles indicate the 'snippets' calculated from the segment and the associated frequency contour. Small red rectangles indicate snippets that were eventually discarded (see below). Small magenta rectangle is the snippet highlighted in panel **E**. (E) Correlation coefficient maps determined from each microphone pair. In each map, the color represents the correlation coefficient between the two microphone signals once each is time-shifted appropriately for that position. Thus, deep blue/ red points represent likely/unlikely source locations, given the information just in this snippet, for just this microphone pair. Plus symbol (+) represents the source location eventually estimated from this individual snippet (see below). Mouse icon represents mouse location. Inset is an enlargement of the area indicated in the upper left map, to show closely intercalated red and blue bands. Map boundaries correspond to the floor outline indicated in **A**. (F) Reduced steered response power (RSRP) map for the example snippet. Plus symbol (+) represents the location estimate for this snippet, and corresponds to the highest (positive) value in the map. Black boundary corresponds to the floor outline in **A**, and microphone locations are indicated by numbered microphone symbols. (G) Consensus estimate from all snippets. Plus symbols (+) and open circles represent single-snippet estimates from all snippets for this segment. Open circles are those snippets determined to be outliers, and non-outlier snippets are pluses. Closed circle indicates the mean of the non-outlier estimates. Gray shading is the probability density of a Gaussian distribution with the mean and covariance matrix of the non-outlier estimates. Mouse probability index value that the vocalization came from the actual mouse was determined to be approximately 1, and from three randomly located virtual mice (gray mouse icons) were $10^{-11}$, $10^{-56}$, and $10^{-75}$. To generate three virtual mouse positions, we picked three random points within the floor of the cage. Black boundary corresponds to the floor outline in **A**, and microphone locations are indicated by numbered microphone symbols.

between the location of the actual mouse and estimated sound source was 3.87 cm (*Figure 2C*). 34.8% of the vocal signals were localized within 3 cm of the mouse and 63.0% were localized within 5 cm (*Figure 2C*; inset). When the heads of two mice are in close proximity, one would expect an increase in type-2 errors (i.e., assigning the vocalization to the incorrect or non-vocalizing mouse). We therefore examined the distance between the mouse assigned the vocalization (mouse 1) and the animal closest to the assigned mouse (mouse 2). *Figure 2D* shows that mouse 1 and mouse 2 were rarely close together (<3 cm) when assigning a vocalization. These results indicate that our system is precise and, of utmost importance, accurate when a vocalization is assigned to a mouse.

Next, we examined the relative position of the estimated sound source locations and the actual mouse position. This was achieved by translating the coordinates of each sound source estimate and the position of the mouse into a mouse-centered reference frame. The results revealed that the estimated sound source locations were clustered around the head of the mouse (*Figure 2E–F*). To quantify these results, we examined an area surrounding the real mouse that was 225 cm$^2$ and centered on the mouse's nose. The large region of interest was then divided into 9 equivalently sized smaller regions (5 cm × 5 cm squares). Region 5 was centered on the mouse's nose and the other 8 regions were located along the periphery of this square (*Figure 2E*). The distribution of predicted sound source locations was significantly different between regions (*Figure 2F*; $\chi^2_{0.05,8} = 2305.7$, $p < 10^{-5}$). Based on a uniform distribution, there were more estimated sound source locations in region 5 than expected, whereas the peripheral regions had fewer. These results indicate that the system precisely and accurately estimates the location of the sound source.

Ultrasonic vocalizations are emitted when two mice approach each other; however, the majority of vocalizations are produced when the animals are in close proximity (*Sewell, 1972*; *Portfors and Perkel, 2014*). To control for this behavioral phenomenon, we ran a second more restrictive simulation. Instead of using 3 virtual mice at random locations within the cage, the simulation randomly generated the position of a single virtual mouse that was located within a 10 cm radius of the true source. Since vocalizations occur across a range of relative distances between two mice, this control analysis will overestimate the error rate and provide a lower bound for the accuracy. When applying our inclusion criteria, the vocal signal was assigned to the real mouse 89.5% of the time and 40.4% of the localized signals were assigned. The decreased assignment rate indicates the

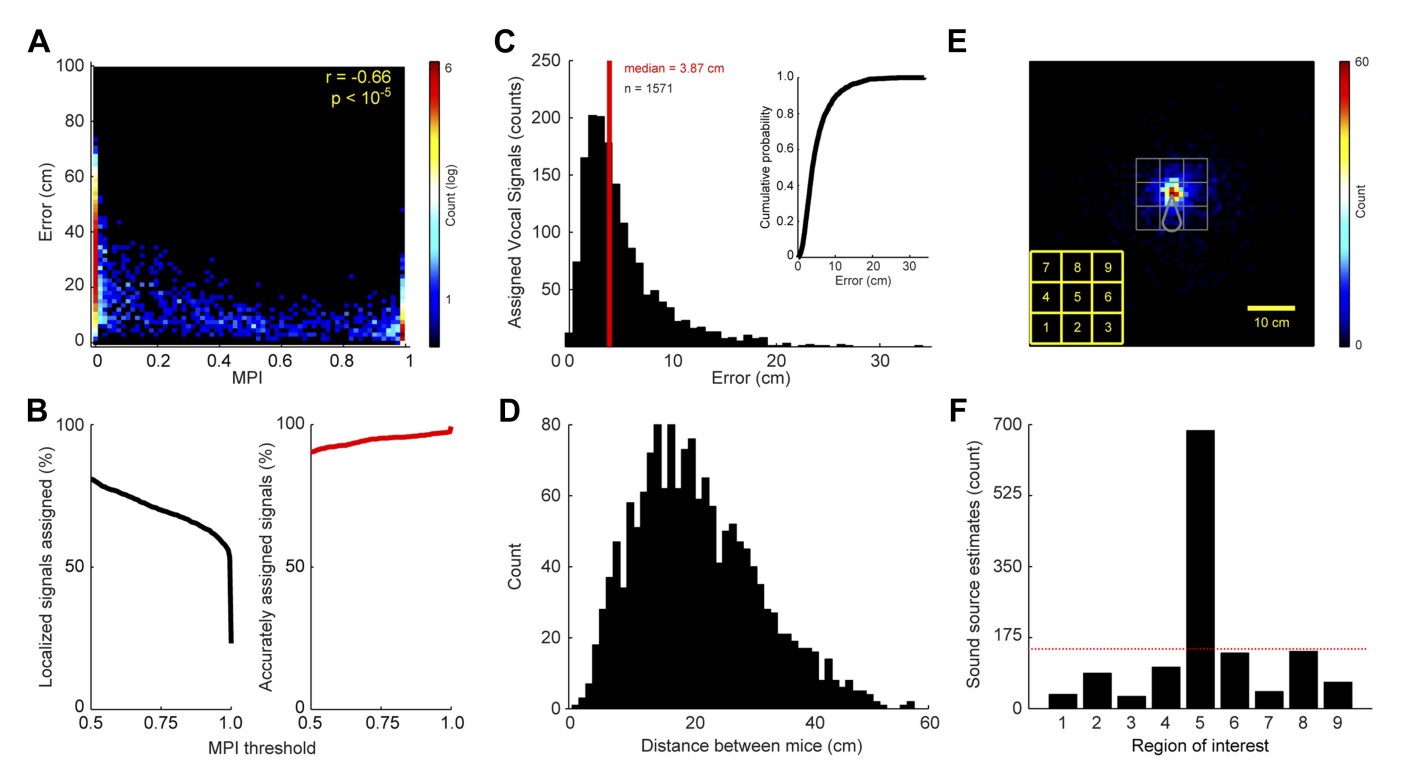

**Figure 2**. Accuracy and precision of sound-source localization system. (**A**) Heat map shows inverse relationship between MPI and the distance between the mouse and estimated sound source location (r = -0.66; p < 10-5). The heat map is plotted in log counts with red indicating the maximum. Data for real and virtual mice are included in the plot. (**B**) Left panel shows the MPI threshold plotted relative to the percent of localized signals that were assigned. Right panel indicates the percentage of accurately assigned vocal signals plotted as a function of MPI threshold. (**C**) Distribution of the errors between the location of the estimated sound source and real source for assigned vocalizations. Median error is plotted in red. Inset shows the cumulative probability histogram of the errors. (**D**) Distribution of distances between the mouse assigned the vocalization and the animal closest to the assigned mouse. (**E**) Heat map showing sound source estimates (n = 2590) relative to mouse position (nose shifted to origin and rotated upwards). Gray sector shows mouse body. The large gray square, which is divided into smaller squares, indicates the regions of interest used to determine the precision of sound source localization system. Yellow key located at the bottom left indicates numbering scheme for smaller regions of interest. (**F**) Bar plot showing the number of sound source localization estimates in each of the smaller regions of interest. Region 5 includes the head of the mouse. Red line shows the expected counts based on a uniform distribution. The distribution of predicted sound source locations was significantly different between regions ($\chi^2_{0.05,8}$ = 2305.7, p < $10^{-5}$).

conservative approach we used when assigning the signals to mice. Consequently, the system allows us to precisely distinguish the sex-specific vocal and physical contributions of each individual mouse during complex social interactions.

## Vocalizations during complex social interactions

The vocal contributions of male and female mice were examined in conditions that increase the complexity of vocalizations and social interactions. Groups of mice (7 groups each consisting of two males and two females) were recorded for five continuous hours. Because group housing affects social behavior in males (*Jones and Nowell, 1989*) and females (*König, 1994*), all mice were singly housed for at least 14 days before the experiment in an attempt to maximize the amount of social information exchanged after the animals were introduced to each other. Our system detected a total of 255,396 vocal signals, of which 199,288 were localized to a distinct position in the cage. The localized signals were further refined to 37,371 signals that could be unambiguously assigned to individual mice (*Figure 3*). *Figure 3A* shows an example of a signal assigned to one of the male mice that was spatially isolated from the other mice in the cage. The 47.1 ms signal was partitioned into 25 snippets and the estimated position of each snippet was located near the tip of the nose of the solitary animal, suggesting that the signal was emitted from that male mouse. *Figure 3B–C* shows additional

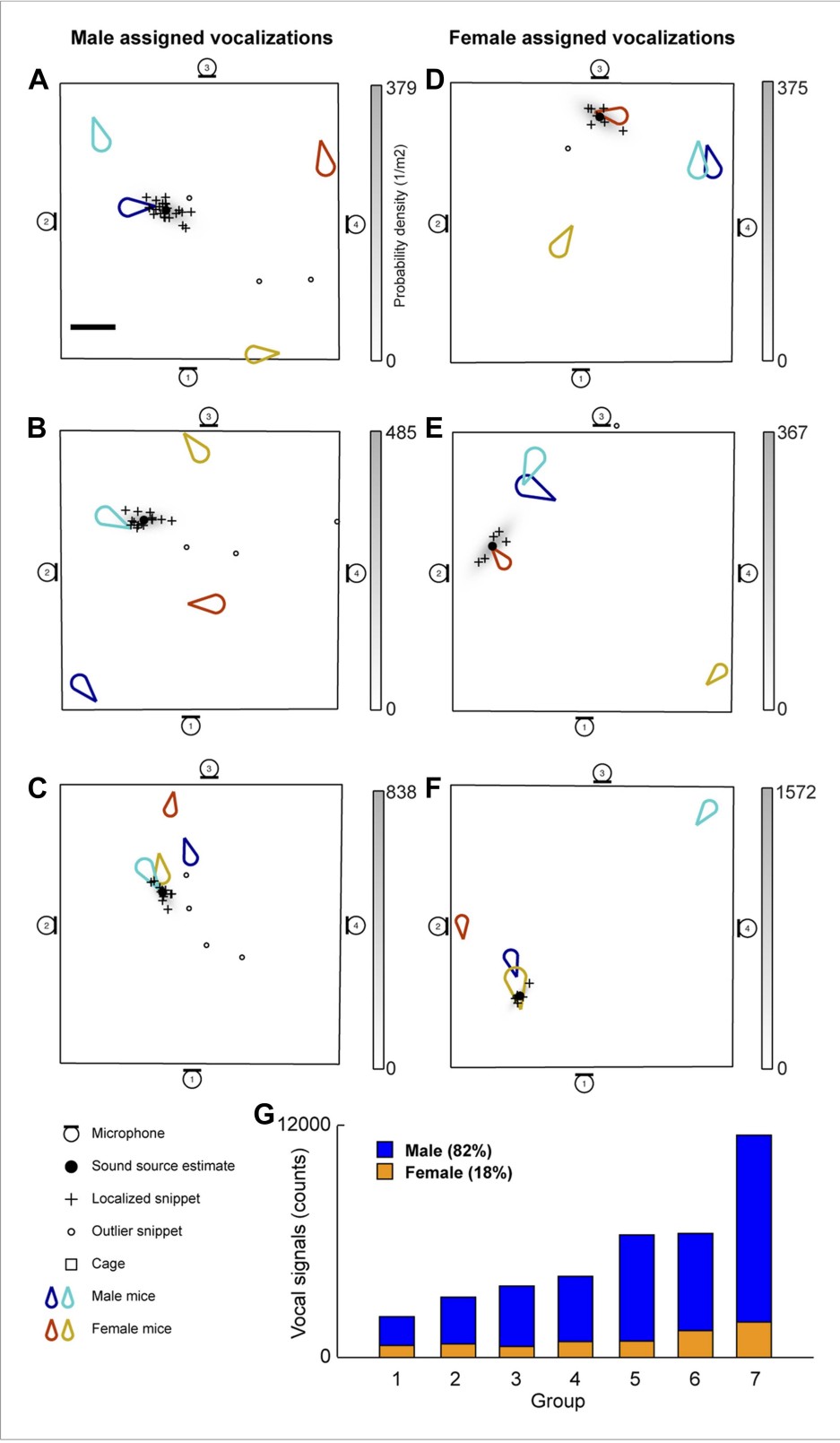

**Figure 3**. Microphone array reveals both male and female mice vocalize during social interactions. (**A–C**) Examples of sound source assigned to a male mice in the presence of multiple mice. (**D–F**) Examples of sound source assigned to female mice when groups of mice are present. (**G**) Male (blue) and female (orange) vocal signal counts during each recording session (groups displayed in ascending order of total call number).

examples of vocal signals that were assigned to a male mouse. Note that the nose of the vocalizing male in *Figure 3C* is in close proximity to the anogenital region of a female. Supporting previous work (*Sewell, 1972*; *Nyby, 1983*; *Chabout et al., 2012*), we directly showed that male mice vocalize in the presence of freely moving female mice.

Males were not the only sex emitting vocalizations. Female mice also vocalized when freely moving male mice were present (*Figure 3D–F*). *Figure 3D* depicts a sound source localization example for a 17.6 ms vocal signal. The signal was partitioned into 8 snippets that were confined to the nose of a female mouse while the other three mice were positioned in remote locations in the cage. As shown in *Figure 3F*, vocal signals can be assigned to a female mouse when a male is positioned behind the female. When combining the data from all the recording sessions, we observed that females produced 18% of the assigned vocalizations (*Figure 3G*), indicating that both males and females vocally contribute while in the presence of the opposite sex.

A possibility exists that the sound source localization system introduces a selection bias because the majority of vocalizations are unassigned. To address this possibility, we examined whether any consistent differences were detected between assigned and unassigned vocalizations as a function of time or space (*Figure 4*). In all groups, both assigned and unassigned vocalizations occurred throughout the experiment (*Figure 4A*). Moreover, the spatial distribution was similar for assigned and unassigned vocalizations (*Figure 4B*). The evidence from these analyses demonstrates that there is no temporal or spatial bias in vocalization assignment. We next examined the relative positions of the two mice closest to the estimated sound source. For all vocal signals, two distances were calculated: (1) the distance between the estimated sound source position and the closest mouse and (2) the distance between the estimated sound source position and the next closest mouse. These two distances were plotted in relation to each other to create two distance distribution matrices, one for assigned and one for unassigned vocal signals. Next, a difference distribution was calculated (assigned-unassigned). The results revealed that assigned vocalizations occurred predominantly when the estimated sound source is close to one mouse, whereas unassigned vocalizations occur when two mice are equidistant from the estimated sound source (*Figure 4C*). This indicates that the major reason for excluding vocalizations was ambiguity in the source of the signal and highlights the effort taken to avoid incorrect assignments. With this system, we were able to follow the vocal and physical interactions of mice in large, mixed sex groups and precisely distinguish sex-specific contributions during social interactions.

Since both sexes vocalize in mixed sex groups, we examined the timing between male and female vocalizations. To explore the temporal relationship between male and female vocalizations, we plotted the vocalizations of both males within +/− 30 s of each female vocalization (*Figure 5B*). *Figure 5C* shows a similar analysis where the vocalizations of both females were plotted relative to each male vocalization. A striking temporal correlation was revealed—when male vocalizations occurred, the probability of female vocalizations peaked. Similarly, when female vocalizations occurred, the probability of male vocalizations peaked. We quantified this effect by calculating the average rate of vocalizations triggered by either sex (*Figure 5D–E*; peak male vocalization rate was 0.78 Hz; one-way ANOVA, $F_{59,360} = 2.04$, $p < 5 \times 10^{-5}$; peak female vocalization rate was 0.19 Hz; one-way ANOVA, $F_{59,360} = 5.2$, $p < 10^{-22}$). On average, both sexes showed an increase in vocalization rate that was significantly higher than expected by chance (random permutation test, $p < 0.006$ and $p < 0.004$). Overall, we found that 24% of male vocalizations occurred within 1 s of a female vocalization, while 61% of female vocalizations occurred within 1 s of a male vocalization. This coordination of vocal behavior may be indicative of an exchange of social information between males and females.

To determine whether this temporal coordination in vocalization was associated with a particular social behavior we identified instances in which a vocalization from a single male was followed within 1 s by one or more vocalizations from a single female. This pattern was called a *vocal sequence* and the vocalizing male and female were called the *vocal pair*. Because four mice were present in the cage, the other mice were called the *non-vocal pair*. For each pair, we measured the speed and distance between the male and female during a 60 s window centered on vocal sequence onset (see *Figure 6A* for example trajectories). The speed of the vocal pair was significantly faster at the time of the vocal sequence than either before or after the event (*Figure 6B*; before: 1–30 s, after: 2–30 s; one-way ANOVA, $F_{1740,10,446} = 11.9$, $p < 10^{-6}$). Despite a slight peak, the speed of the non-vocal pair did not change significantly over time (one-way ANOVA, $F_{1740,10,446} = 1.0$, $p > 0.9$). We believe this peak

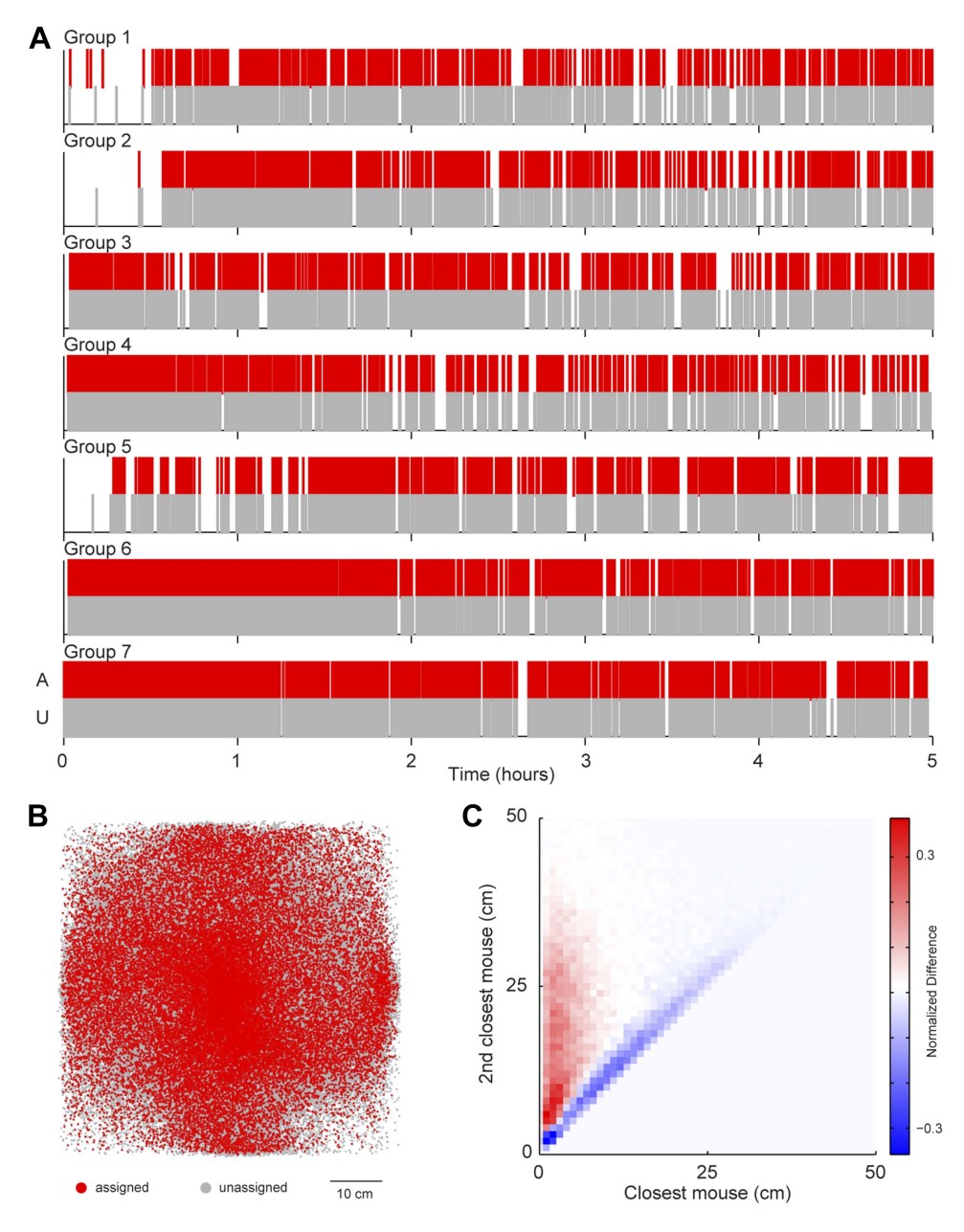

**Figure 4**. Potential sources of bias in vocalization assignment. (**A**) Assigned (red) and unassigned (gray) vocalizations are plotted as a function of time. In all groups, assigned and unassigned vocalizations occur throughout the experiment. (**B**) The estimated sound source position of assigned (red) and unassigned (gray) vocalizations is displayed. Both assigned and unassigned vocalizations are less likely to be detected in the corners, but there is no difference in the spatial distribution for the two categories. (**C**) The relative distance between sound source and the closest mouse (x-axis) or the second closest mouse (y-axis) for all vocalizations. Relative distances were calculated separately for both assigned and unassigned vocalizations, a 2-dimensional distance matrix was determined for each category and normalized by peak count, and then the difference between the distance distribution matrix for assigned and unassigned vocalizations was plotted. Red represents a higher proportion of assigned vocalizations, whereas blue denotes a higher proportion of unassigned vocalizations. Vocal signals were unlikely to be assigned when the closest and second closest mouse were equidistant from the estimated sound source (blue diagonal line).

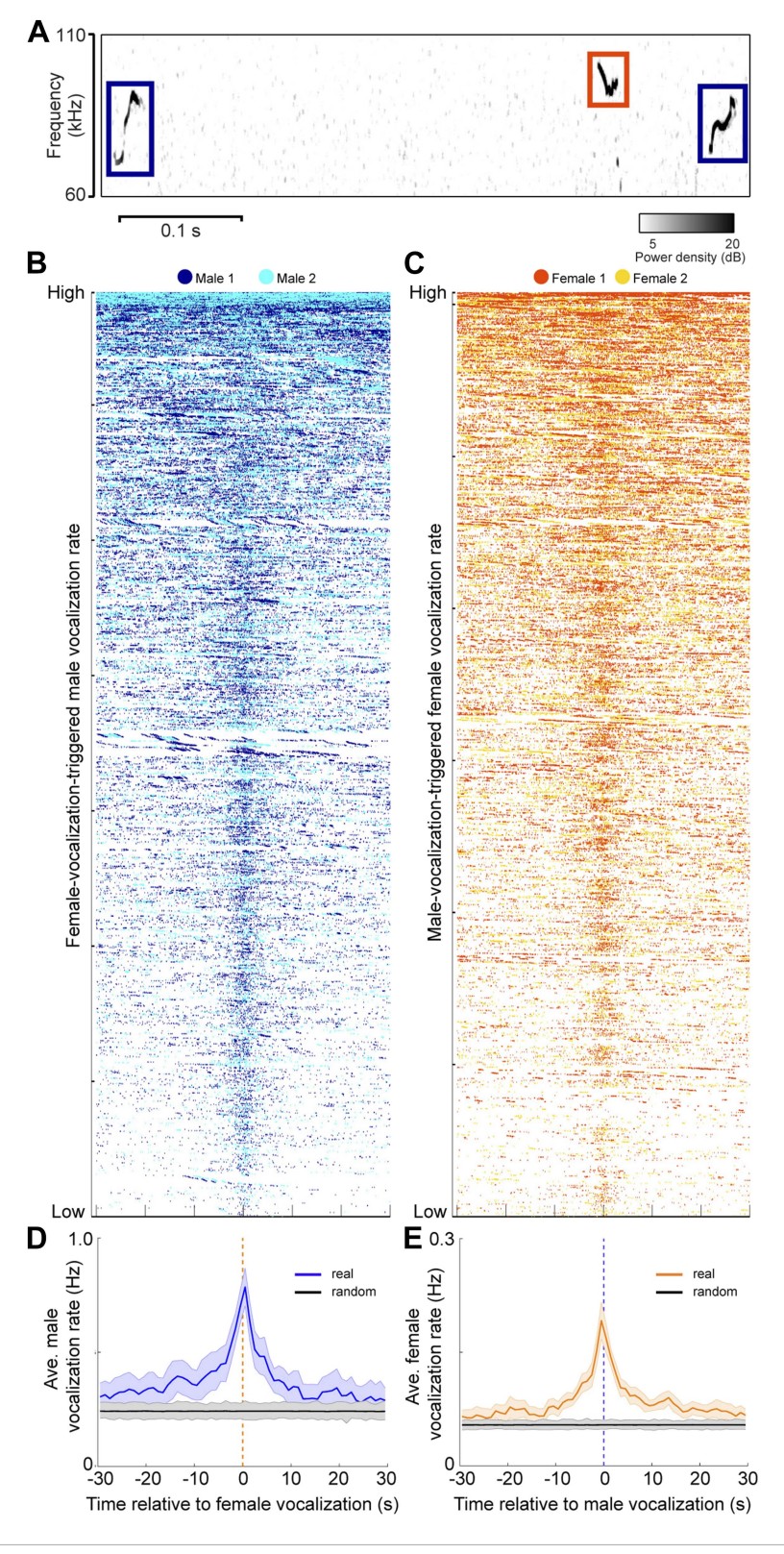

**Figure 5**. Male and female mice vocalize together. (**A**) Example spectrogram of male and female USVs. (**B**) Raster plots of male USVs plotted relative to female vocalizations (each row is associated with a single female vocalization with onset at t = 0 s; female vocalizations = 6832; rows sorted by male vocalization rate). (**C**) Female USVs plotted relative to male vocalizations (male vocalizations = 30,395; rows sorted by female vocalization rate). (**D–E**) Plots of

*Figure 5. continued on next page*

*Figure 5. Continued*

female-vocalization-triggered average male USV rate (**D**) and male-vocalization-triggered average female USV rate (**E**). Colored lines represent group averages (n = 7) with SEM (shaded patch). Black lines represent group averages (n = 1000) for randomly generated trigger times. Light gray patches show the range of the randomly generated group averages.

occurs because all four mice are in the same cage, and therefore the behavior of the vocal pair may affect the behavior of the non-vocal pair. Moreover, the members of a vocal pair were significantly closer to each other at the onset of the vocal sequence than either before or after the event (*Figure 6C*; before: 30–3.2 s, after: 5–30 s; one-way ANOVA, $F_{1740,10,446} = 15.5$, p < $10^{-6}$). In contrast, the distance between non-vocal mice did not change significantly over time (one-way ANOVA, $F_{1740,10,446} = 0.4$, p > 0.99). For the vocal pair, we examined the relative positions of the mice before, during, and after the vocal sequence (−25, 0, and 25 s, respectively). At the time of the vocal sequence, the male positions were significantly more clustered behind the females than either before or after the event (*Figure 6D*; female-centered: $\chi^2_{0.05,2} = 220.9$, p < $10^{-5}$, Tukey-type multiple comparison, before, $q = 15.1$, p < 0.001, after, $q = 16.9$, p < 0.001; male-centered: $\chi^2_{0.05,2} = 198.9$, p < $10^{-5}$, Tukey-type multiple comparison, before, $q = 15.1$, p < 0.001, after, $q = 14.4$, p < 0.001). These analyses show that vocal interaction between a male and female can occur when the mice are in close proximity to each other, moving quickly, with the male positioned behind the female. This behavior strongly resembles a chase, a fundamental component of mouse courtship (*Van Oortmerssen, 1971*).

Our previous behavioral analyses were based on epochs when the initial trigger vocalization was from a male; however, the behavior of the vocally interacting animals may be different when the female starts the sequence. To investigate this possibility, additional analyses were performed for vocal sequences initiated by females, revealing similar results. Vocally interacting animals moved the fastest at the time of the vocal sequence (before: 30–0.9 s, after: 2.5–30 s; one-way ANOVA, $F_{1740,10,446} = 14.3$, p < $10^{-6}$). For the non-vocal pair, the speed remained relatively constant (one-way ANOVA, $F_{1740,10,446} = 0.7$, p > 0.99). Furthermore, as in the male-initiated vocal sequences, the vocal pair was significantly closer to each other at the vocal sequence onset than either before or after the vocal exchange (before: 30–3.2 s, after: 3.6–30 s; ANOVA, $F_{1740,10,446} = 16.4$, p < $10^{-6}$). On average, non-vocal mice remained 35 cm apart (ANOVA, $F_{1740,10,446} = 0.3$, p > 0.99). The positions of the males were significantly more clustered behind the females at the time of the vocal sequence compared to before or after the event (female-centered: $\chi^2_{0.05,2} = 135.0$, p < $10^{-5}$, Tukey-type multiple comparison, 25 s before, $q = 9.9$, p < 0.001, 25 s after, $q = 14.4$, p < 0.001; male-centered: $\chi^2_{0.05,2} = 178.1$, p < $10^{-5}$, Tukey-type multiple comparison, 25 s before, $q = 15.1$, p < 0.001, 25 s after, $q = 11.9$, p < 0.001). These results indicate that both male and female initiated vocal sequences occur during similar behaviors.

To identify when males were chasing females, we manually annotated a subset of chases and used this annotation to train an automatic classifier (*Kabra et al. (2013)*; see Materials and methods). In every cage, each male chased each female (median = 113; interquartile range (IQR) = 162.5–35). In addition, each male vocally interacted with each female (median = 111; IQR = 166–77.5). The relationship between chases and vocal interactions was then examined for all male and female pairs (n = 28; *Figure 7A*). The proportion of chases performed by each male varied across groups. In some cases, only one of the males did the majority of the chasing (e.g., group 4, red symbols). In other cases, both males chased equally (e.g., group 6, pink symbols). Similarly, the amount each of the females was chased varied. Most of the time, one of the females in the group was chased more than the other (e.g., groups 1, 3, 4, 6, and 7; black, green, red, pink, and cyan symbols). In contrast, females in groups 2 and 5 were chased for similar amounts of time (blue and gray symbols). Despite this variation, the number of male-initiated vocal interactions was strongly correlated with both total chase time (*Figure 7A*, r = 0.74, p < $10^{-5}$) and number of chases (r = 0.55, p < 0.003). Given this correlation we asked whether there was a consistent temporal relationship between vocal interactions and chases. Vocal interactions that occurred within 30 s of a chase (1743/3075) were plotted in relation to the onset of the nearest chase (*Figure 7B*). The rate of vocal interactions occurring inside a chase was significantly higher than outside the chase (inside: median = 0.18 Hz, IQR = 0.61–0; outside: median = 0.02 Hz, IQR = 0.04–0; Mann–Whitney U-test, z = 9.74, p < $10^{-21}$), with 47.2% of

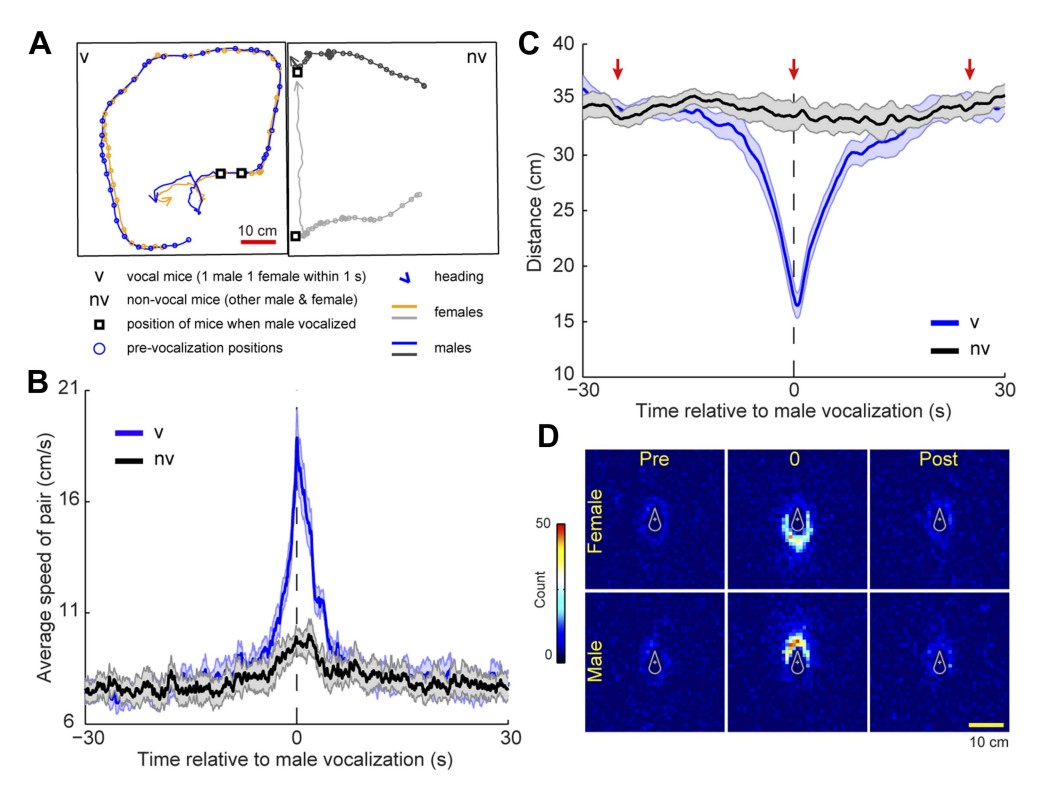

**Figure 6**. Mice participating in a vocal sequence are close to each other. (**A**) Example trajectories of vocal and non-vocal pairs of mice. (**B**) Vocal-sequence-triggered averages show that vocal (v) mice are faster at the time of the initial vocalization than either before or after the event (p < 10⁻⁶). The speed of the non-vocal (nv) mice did not change significantly over time (p > 0.99). For each event, the instantaneous speeds (±30 s from trigger start time) were averaged for the vocal or non-vocal pair of mice. Colored and black lines represent the average speed between vocal and non-vocal pairs, respectively. (**C**) The vocal pair was significantly closer at vocal sequence onset than either before or after the event (p < 10⁻⁶). Average distance between non-vocal mice did not change significantly (p > 0.99). Red arrows denote the periods in panel **D**. For **B** and **C**, shaded patch indicates SEM across groups (n = 7) and dashed vertical lines denote the time of initial trigger vocalizations. (**D**) Heat maps of relative position of female (top row) and male (bottom row) mice preceding (pre), during (0), and following (post) a vocal sequence. Males are significantly clustered behind the females at vocal sequence onset compared to before or after (p < 0.001).

these vocal interactions occurring during a chase (823/1743). Analyses of female-initiated vocal interactions showed a similar pattern (total chase time–vocal interaction correlation: n = 28; r = 0.80, p < 10⁻⁶; vocal interaction and chase timing: inside: median = 0.16, IQR = 0.50–0; outside: median = 0.02, IQR = 0.02–0; Mann–Whitney U-test, z = 9.77, p < 10⁻²¹). Our observation that the vocal interaction rate increased during chases suggests that both males and females actively participate in courtship displays.

Chase-related vocal interactions occurred when animals were in close proximity. Therefore, a possibility exists that the female's vocal contribution to the interaction was actually a male vocalization incorrectly assigned to a female. To control for this possibility, we used the data from the single mouse recordings and simulated a close proximity social interaction by randomly generating a second virtual mouse. To determine the possible positions of the virtual mouse that reflected the relative positions of vocally interacting mice in the multiple mouse experiment, we calculated the distance between each vocally interacting male and female at the time of the female vocalization as well as the orientation of the female relative to the male. The distance and orientation distributions were then sampled with replacement to generate the position of the virtual mouse relative to the real mouse in the single mouse experiments. In this control simulation, vocal signals were assigned to the real mouse 93.0% of the time, which indicates a 7% error rate.

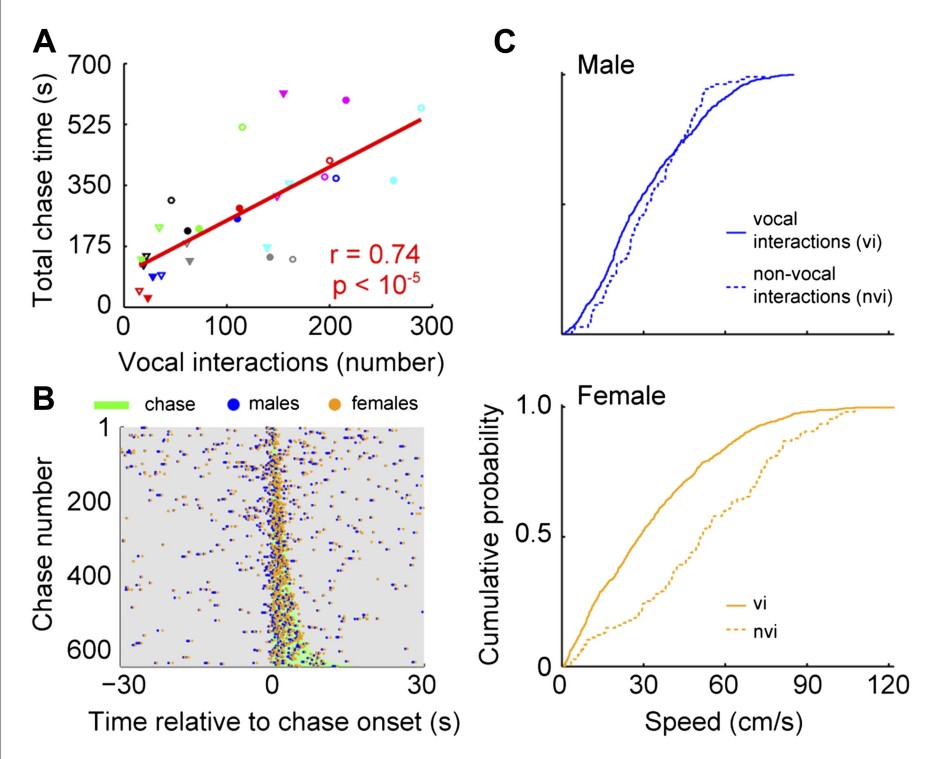

**Figure 7.** Vocal interactions are associated with courtship. (**A**) The number of male-vocalization-initiated vocal interactions and total chase time for a given male-female pair was strongly correlated (n = 28; r = 0.74, p < 10$^{-5}$). Trend line (red line) was calculated using linear regression. Circles and triangles represent male 1 and male 2, respectively. Open and filled depict female 1 and female 2, respectively. Colors indicate group numbers (black = 1, blue = 2, green = 3, red = 4, gray = 5, pink = 6, and cyan = 7). (**B**) Plots of male and female USVs as a function of time relative to chase. Vocal interaction rate is higher during chases than outside chases for male-initiated vocal interactions (p < 10$^{-21}$). (**C**) Mouse speeds at the time of the male vocalization during chases with and without vocal interactions. Top, male speed (p > 0.25). Bottom, female speed (p < 10$^{-10}$).

A series of analytical calculations were performed to determine the number of expected female vocalizations and vocal interactions. In the 1743 vocal interactions that occurred within 30 s of a chase, there were a total of 3486 vocalizations emitted. If male mice were the only animals vocalizing and all female vocalizations were incorrectly assigned based on a 7% error rate, we would have only expected a total of 244 female vocalizations. Instead, 1743 vocalizations were assigned to the female, which is greater than 7 times the number of expected incorrectly assigned male vocalizations. This analysis not only suggests that female mice vocalize in the presence of males, but moreover, they are vocally interacting with the males during courtship.

Because female mice vocally participate in courtship chases, we investigated the specific role of the female's vocalization during vocal interactions. To isolate the effect of the female vocalization, we compared chases in which both the male and female vocalized (vocal interaction chases) with those in which only the male vocalized (male only chases). We used speed at the time of the male vocalization as a proxy for female receptivity (*Kowalski et al., 2004*). When examining the speed of the males, we found no difference between the two chase types (*Figure 7C*, top panel; vocal interaction: median = 28 cm/s, IQR = 0.45–0.17; non-vocal interaction: median = 31 cm/s, IQR = 0.45–0.20; Mann–Whitney U-test, z = 1.13, p > 0.25). In contrast, female speed was significantly slower during chases with vocal interactions than without vocal interactions (*Figure 7C*, bottom panel; vocal interaction: median = 28 cm/s, IQR = 0.49–0.12; non-vocal interaction: median = 51 cm/s, IQR = 0.73–0.32; Mann–Whitney U-test, z = 6.73, p < 10$^{-10}$). In addition, vocal interaction chases were on average longer in duration than non-vocal interaction chases (vocal interaction: median = 3.28 s, IQR = 5.02–2.07; non-vocal interaction: median = 2.38 s, IQR = 3.72–1.22; Mann–Whitney U-test, z = 4.45,

$p < 10^{-5}$). Our results show that the female drives the overall reduction in speed during a chase; thus, female vocal interaction may function as a signal of receptivity.

## Discussion

In this study, we used a novel microphone array system to reveal an unprecedented level of vocal exchange between male and female mice during courtship. In the discussion, we consider and expand upon the most salient components of social vocalizations that emerged from our investigations, focusing on the potential impact of the tool, the role of female vocalizations, the synchrony of male and female vocalizations, and the function of vocal interactions during courtship.

Mice produce ultrasonic vocalizations in many social contexts including establishment of territory, resident–intruder interactions, pup care, juvenile play, social reunion, and mating (*Sales and Pye, 1974*). In all of these social contexts, the presence of multiple vocalizing mice has hindered our ability to determine the specific role vocalizations play in social behaviors. Assigning a vocalization to a particular mouse in a social group is difficult for two reasons. First, there is no obvious visual sign that a mouse is producing a USV (*Chabout et al., 2012*). Second, although mice appear to produce characteristic vocal patterns or songs (*Holy and Guo, 2005*; *Sugimoto et al., 2011*; *Hoffmann et al., 2012*), individual ultrasonic vocalizations are similar across animals, both within and between sexes (*Hammerschmidt et al., 2012a*). Therefore, a vocalization's acoustic features are likely inadequate for identifying which mouse in a social group emitted the sound. Previous studies have addressed the difficulty in determining the source by assuming a single mouse produced the majority of the observed vocalizations (*Barthelemy et al., 2004*; *Choi et al., 2011*; *Hanson and Hurley, 2012*), or by experimentally preventing one mouse from vocalizing (*Whitney et al., 1973*; *Warburton et al., 1989*). Both conditions overlook how social interactions may be impacted by the vocal contribution of multiple individuals. More importantly, these conditions cannot detect vocal exchanges between individuals. Consequently, the ability to track the vocal behavior of each animal is crucial in determining the function of mouse USVs. To examine the vocal contribution of every mouse during a social interaction, we developed and employed a microphone array system. In the past, microphone arrays have been used to study the acoustic features of many species (*Payne et al., 2003*; *Mennill et al., 2006*; *Chiu et al., 2010*; *Collier et al., 2010*; *Kalcounis-Rueppell et al., 2010*). Other systems capitalized on the power of an array to determine the number of animals hidden in dense terrain (*Celis-Murillo et al., 2009*; *Blumstein et al., 2011*) or map the boundaries of an animal's territory (*Kirschel et al., 2011*), whereas our system was specifically designed to link, with high resolution and accuracy, the vocal and social behaviors of mice recorded in a laboratory environment.

Group size influences the behavior of individual animals (*Fitzsimmons and Bertram, 2013*; *Shen et al., 2014*). For example, in a variety of species, the full range of a male's vocal repertoire only occurs when multiple males are present (*Rand and Ryan, 1981*; *Martinez-Rivera and Gerhardt, 2008*; *Charlton and Reby, 2011*). Moreover, recent work has shown that mouse vocal behavior (*Portfors, 2007*) and patterns of social interactions (*Shemesh et al., 2013*; *Weissbrod et al., 2013*) are significantly different when mice are housed in more complex social groups than in dyads. These studies emphasize the importance of capturing the full behavioral repertoire. In the present study using a more dynamic social environment of two males and two females, we show that all mice vocalize and vocal exchanges occur between all possible male-female pairs. These findings illustrate the utility of the microphone array for clarifying the function of vocalizations in mouse social behavior.

One of the most remarkable discoveries from the study was that male and female mice coordinate their ultrasonic vocalizations during courtship. These results are surprising because most previous studies concluded that only male mice produce courtship USVs (*Whitney et al., 1973*; *Warburton et al., 1989*; *Barthelemy et al., 2004*). However, male and female mice emit USVs with practically identical acoustic structure (*Hammerschmidt et al., 2012a*), leaving open the possibility that some courtship USVs attributed to males are actually produced by females. Moreover, it is known that female mice possess the neural circuitry necessary to emit USVs during interactions with males because surgical or genetic lesions to the vomeronasal organ unmask this behavior (*Kimchi et al., 2007*). Our findings expand on this observation and show that the behavior can be triggered in unaltered females. Our observation that the majority of a female's vocalizations occur during vocal exchanges with males suggests that female vocal behavior may depend on the presence of a vocally competent partner. This in turn, may explain the absence of ultrasonic vocalizations when female mice are exposed to male odor (*Maggio and Whitney, 1985*) or paired with an anesthetized (*Whitney et al., 1973*) or

devocalized (*Warburton et al., 1989*) male. Taken together, these results suggest that information from multiple modalities is necessary to activate these social neural networks.

For courtship displays, the behavior of both partners is critical. Female responsiveness modulates male mating behavior throughout the animal kingdom (*Coleman et al., 2004*; *Higham et al., 2009*; *Akre and Ryan, 2011*). For example, in *Drosophila melanogaster*, males modulate their song patterning in response to variation in sensory cues about the female's position and motion (*Coen et al., 2014*). Without accounting for the behavior of the female, the male's song pattern appears random. A similar interdependence of behavior between partners occurs in the early stages of human courtship, where a female's non-verbal displays modulate a male's approach behavior (*Moore, 1985*). Interactions between males and females during courtship also occur in the vocal domain (*Janson, 1984*; *Tobias et al., 1998*; *Garamszegi et al., 2007*). Most studies have found that vocal behavior is either important for joint territorial defense or maintaining affiliation, particularly between mated pairs (*Hall, 2004*). The association between vocal interactions and courtship chases in this study implies that vocal exchanges between mice may be involved in affiliation. As for the role of female vocalizations in courtship, many believe a female's participation may serve as an indication of her receptivity (*Janson, 1984*) or may encourage competition between males (*Montgomerie and Thornhill, 1989*). In rats (*Thomas and Barfield, 1985*) and hamsters (*Floody et al., 1977*), female vocal participation in courtship is modulated by estrous state and has been proposed to signal female receptivity to nearby males, also consistent with a role in affiliation. Our results strengthen the argument that female vocalizations are associated with receptivity in rodents, since female speed during a chase is significantly slower when she subsequently vocalized than when she was silent. This is akin to other species; receptive females slow down permitting the courting male to approach and mate (*McGill, 1962*; *Beach, 1976*; *Hall, 1994*; *Kowalski et al., 2004*; *Szykman et al., 2007*). Based on the behaviors of the animals that co-occur during these vocal exchanges, we believe that the female's vocal contribution may facilitate this bond by indicating her receptivity.

In many species, social interactions are essential for reproductive success and survival (*Bradbury and Vehrencamp, 1998*). These interactions are supported by information exchange through a variety of sensory modalities (*Adolphs, 2010*). Transfer of social information occurs over a wide range of timescales, from short-term information about an individual's current motivational state (*Moles and D'Amato F, 2000*) to longer-term information about dominance status (*Ficken et al., 1987*; *Grosenick et al., 2007*) or fertility (*Leong et al., 2003*). Capturing the details of this information exchange is critical to developing a mechanistic understanding of how social information supports motivated behavior, and also in illuminating deficits of social interaction, such as autism. The microphone array system described here allows unprecedented access to the details of social interactions in groups of freely behaving mice, revealing temporally precise and behaviorally meaningful vocal communication between males and females.

## Materials and methods

### Subjects

Singly housed male mice (C57Bl/6J; n = 3; 3–5 months) were used to test the sound source localization system. For multiple mouse studies, two male and two female mice were used in each experiment (C57Bl/6J; n = 28; 6-12 weeks). Mice were isolate housed for at least two weeks prior to the start of the recordings and maintained on a 12/12 dark–light cycle with *ad libitum* access to food and water.

At least one week prior to the start of the recordings, mice were marked with distinctive patterns for identification by applying harmless hair bleach to the fur (*Ohayon et al., 2013*). The patterns were two vertical lines, two horizontal lines, one diagonal slash, or five dots. Each of the four subjects randomly received one of the four patterns. In rodents, reports indicate that USVs are important for female receptivity during estrus (*Floody et al., 1977*; *McIntosh et al., 1978*) and, therefore two hours prior to the anticipated start of the experiment, non-invasive lavage and cytological assessment of vaginal cells were performed. Briefly, cells were collected by washing with 20 μl of sterile saline, placed on a slide, stained with Wright Stain, and examined under a light microscope. As described by *Karim et al. (2003)*, estrous stage was calculated based on the proportion of cell types observed. Proestrus consisted of mostly nucleated basal epithelial cells. In estrus, most cells were cornified squamous epithelial cells that lacked a nucleus. During metestrus, cells were a mixture of neutrophils and cornified squamous epithelial cells. Diestrus consisted of mostly neutrophils. If both females were

in late proestrus/early estrus recordings were conducted. Otherwise recordings were postponed and the procedure for determining estrus was repeated again the following day. Male bedding was introduced into the female cage one day before experiment onset to ensure that females were cycling. Experiments began at dark cycle onset and lasted for five hours. Each mouse was individually recorded for three minutes following the experiment. HHMI Janelia Research Campus Institutional Animal Care and Use Committee approved all experimental protocols.

## Recordings

Audio data were captured with a 4-channel microphone array (Avisoft-Bioacoustics, Glienicke, Germany; CM16/CMPA40-5V), amplified (40 dB), low-passed filtered (200 kHz; Krohn-Hite, Brockton, MA; Model 3384), and digitized at 450450 Hz with a National Instruments board (Austin, TX; PXIe-1073, PXIe-6356, BNC-2110). Externally triggered video data were acquired at 29 Hz with a camera (Basler, Ahrensburg, Germany; A622f) and stored on a PC (Dell, Round Rock, TX; T7500) with StreamPix software (Norpix, Montreal, Canada; StreamPix 5). To synchronize the audio and video data, a 29 Hz square wave pulse was emitted from a function generator (Agilent Technologies, Santa Clara, CA; Model 33522A-002). A BNC splitter was used to simultaneously send the output signal from the function generator to the camera and the National Instruments equipment. This signal was time stamped at the sampling rate of the audio recordings and triggered the camera. Custom software was used to control and synchronize the data and video acquisition equipment. Recordings were made in a mesh-walled (McMaster-Carr, Robbinsville, NJ; Nylon) cage, with a frame of extruded aluminum (8020, Inc.; width = 66 cm, length = 66 cm, height = 66 cm), and surrounded with Sonex foam (Pinta Acoustic, Inc., Minneapolis, MN; VLW-35). The cage was illuminated with infrared lights (Reytec imaging, East Setauket, NY; part # IR-LT30) and filled to a depth of ~7.5 cm with alpha-dri bedding (Shepherd Specialty Papers, Richland, MI). A ring of LEDs, which was visible through the mesh walls, surrounded each microphone and helped to determine the microphone positions.

## Tracking

The position of each mouse was automatically tracked using the Motr tracker program (Motr; *Ohayon et al. (2013)*; http://motr.janelia.org). Motr fits an ellipse around each mouse and reports the x and y position of the ellipse centroid, the length of the ellipse's major and minor axis, and the heading direction of each mouse for every frame in the video.

## Audio segmentation

Vocal signals were automatically extracted from the four channels of auditory recording using multi-taper spectral analysis. After removing signals below 30 kHz, overlapping segments in time were Fourier transformed using multiple discrete prolate spheroidal sequences as windowing functions (K = 43, NW = 22). An F-test (*Percival and Waldan, 1993*) was used to infer whether each time-frequency point was significantly above noise based on these independent estimates of intensity (p < 0.05). This procedure was performed for multiple segment lengths on each microphone channel to capture data at different temporal and spectral scales (NFFTs = 128, 256, 512). The data were combined in a single spectrogram whose pixel size corresponded to the time resolution of the shortest segment and frequency resolution of the longest. This image was then convolved with a square box (15 pixels in time X 7 pixels in frequency) to fill in small gaps before the locations of contiguous regions, which exceeded a minimum pixel number (1500), were characterized. Because one or more animals could produce discontinuous vocal signals, each discontinuous signal was extracted separately unless harmonics were present. Overlapping signals were considered harmonics when the overlapped length exceeded 90% of the shortest signal, and the frequencies of 90% or more of the overlap were within 10% of a factor of two or three of each other.

## Preparing data for localization

Each vocal signal was preprocessed to run in the mouse sound source estimation program. The preprocessing steps involved cutting each extracted signal into time-frequency 'snippets'—filtered pieces of the signal 5 ms long and 2 kHz wide. Because video data were collected at 29 Hz (roughly 35 ms), each frame was associated with 7 time bins of audio data. Each of time bins used the same

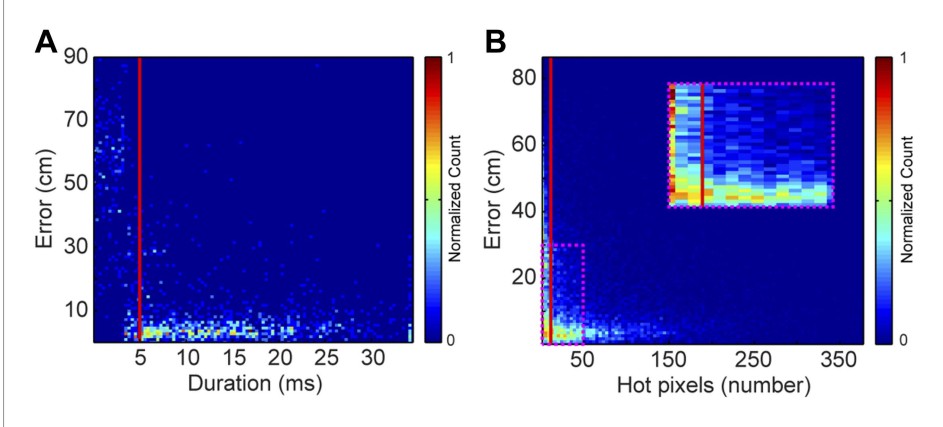

**Figure 8**. Effects of acoustic structure on localization fidelity. (**A**) Localization error as a function of sound duration. The heat map shows that signals shorter than 5 ms (red vertical line) produce large errors between the estimated sound source location and the true position of the mouse. (**B**) Localization error as a function of hot pixel count (pixels in the sound's spectrogram that are significantly above background). Localization error is variable when the hot pixel count is low, and becomes more accurate above 11 hot pixels (red vertical line). Inset (dashed magenta box) shows zoomed in region of heat map highlighting the increase in localization error for snippets with fewer than 11 hot pixels. Heat maps were normalized by peak counts.

mouse position, making the localization sampling rate dependent upon the video sampling rate, as opposed to a per snippet sampling rate of 200 Hz. To determine the optimal duration of snippets, localization error as a function of snippet duration was plotted (*Figure 8A*). Localization error increased dramatically for snippet durations under approximately 4 ms. Consequently, we selected 5 ms for snippet durations to improve the accuracy of the system. To select a frequency bandwidth, we examined 'hot pixels'—pixels in the combined spectrogram that pass the multi-taper F-test for being significantly louder than noise. *Figure 8B* shows that localization error exponentially decays as the numbers of hot pixels within a snippet increases. After applying a cutoff criterion of 11 hot pixels, a bandwidth of 2 kHz was selected because it produced more than three snippets for most vocalizations. Therefore, localization was restricted to snippets that were 5 ms long and contained at least 11 hot pixels. This allowed us to use multiple estimates to determine the source location. Cage boundaries and microphone positions were manually annotated using the images from the video recordings.

## Sound source estimation

Mouse sound source estimation was performed using a procedure similar to that of *Zhang et al. (2008)*. Vocal signals were recorded on an array of four microphones, and a single location was found that best explained the different time delays observed between the six possible combinations of microphone pairs. A grid of points was generated that covered the floor of the arena with a spacing of 0.25 mm, and then the likelihood that the source of a vocal signal could be at a given point on the grid ($\mathbf{r}_g$; 'g' for guess) was determined.

At each point, we computed the distance ($d_k$) from $\mathbf{r}_g$ to microphone $k$,

$$d_k = \|\mathbf{r}_g - \mathbf{r}_k\|, \tag{1}$$

where $\mathbf{r}_k$ is the position of microphone $k$. The signal recorded at microphone $k$ was assumed to be proportional to the true signal, but delayed by a time $\tau_k$ given by:

$$\tau_k = d_k/c, \tag{2}$$

where $c$ is the speed of sound in air at the ambient temperature and standard atmospheric pressure (760 mm Hg).

We then calculated the *steered response power* (SRP), given by:

$$\text{SRP} = \sum_{i=1}^{K} \sum_{k=1}^{K} r_{ik}((\tau_i - \tau_k)/\Delta t), \tag{3}$$

where $r_{ik}(\cdot)$ is the cross-correlation between signals recorded at microphone $i$ and microphone $k$, given by:

$$r_{ik}(p) = \sum_{n=0}^{N-1} V_i[m] V_k^{\star}[m] \exp(j2\pi mp/N), \tag{4}$$

where $V_i[m]$ is the discrete Fourier transform of the voltage recorded at microphone $i$. The SRP is the sum of the cross-correlations between each ordered pair of microphones (including self-pairs), each evaluated at the time difference that would occur if the source was at the hypothesized location.

However, the self-terms in the SRP (where $i = k$) do not vary with position $\mathbf{r}_g$, and because,

$$r_{ik}((\tau_i - \tau_k)/\Delta t) = r_{ki}((\tau_k - \tau_i)/\Delta t), \tag{5}$$

it suffices to calculate the *reduced* SRP, or RSRP, given by:

$$\text{RSRP} = \sum_{i=1}^{K} \sum_{k<i}^{K} r_{ik}((\tau_i - \tau_k)/\Delta t). \tag{6}$$

That is, the RSRP includes exactly one term per unordered pair, not including self-pairs. The RSRP is a function of position, because the hypothesized delays $\tau_i$ and $\tau_k$ depend upon the hypothesized position $\mathbf{r}_g$. We therefore evaluate the RSRP at each grid point $\mathbf{r}_g$, and our estimate of the source location is given by the $\mathbf{r}_g$ that maximizes RSRP. We sped up the computation by pre-computing each $r_{ik}(p)$ for the range of possible $p$ values, and computing $r_{ik}(p)$ at each $\mathbf{r}_g$ by linear interpolation into these pre-computed arrays. The fine spacing of the grid (0.25 mm) was used to enable accurate localization of the maximum of RSRP. Ultrasonic vocal signals have power up to about 130 kHz and at this frequency the wavelength of the sound waves is ~2.6 mm; thus our spacing was approximately 10 times finer than the shortest wavelengths present in the signals. Mouse Ultrasonic Source Estimation software is available for download at https://github.com/JaneliaSciComp/Muse.

## Assigning vocal signals to mice

To assign vocal signals to specific mice, we used the estimated source position in combination with the location of the mice at the time of the vocal signal and a confidence measure in the estimated source position. Location outliers were identified and removed, and the covariance matrix of the remaining snippets was calculated using the method of *Peña and Prieto (2001)*. The x and y coordinates of the non-outlier snippets were then averaged to determined the estimated x and y coordinates of the vocal signal (i.e., estimated source position). The covariance matrix and estimated source position were then used to generate a probability density function over the cage for the vocal signal. The density at each mouse was then calculated from the probability density function. Densities (D) were used to calculate a mouse probability index (MPI) for each mouse using the following formula:

$$MPI_n = \frac{D_n}{\sum_{i=1}^{M} D_i},$$

where $n$ = mouse index and $M$ = total number of mice. Vocal signals were only assigned to a mouse if the *MPI* was greater than 0.95 and the density was greater than 1 m$^{-2}$. By setting the *MPI* threshold to 0.95, vocal signals were not assigned to a mouse when multiple mice were in close proximity to the estimated source. A density threshold of 1 m$^{-2}$ was used to prevent a poorly localized vocal signal from being assigned to a mouse far from the estimated source location. Because of the presence of multiple mice in the cage, overlapping vocalizations were observed. The majority of the time these overlapping vocalizations were unassigned (either the mice were equidistant to estimated source or the snippets from the overlapping vocalizations did not tightly cluster). These vocalizations were excluded from analysis. *Table 1* lists the vocal signals detected, localized and assigned for each data set.

**Table 1.** Vocal Signals detected, localized, and assigned for each data set

| Session | Signals | Localized | Assigned | Individual male 1 | Male 2 | Female 1 | Female 2 | Proportion assigned* |
|---------|---------|-----------|----------|-------------------|--------|----------|----------|----------------------|
| 1 | 23,292 | 17,444 | 2115 | 1021 | 465 | 364 | 265 | 0.12 |
| 2 | 25,942 | 19,531 | 3118 | 1903 | 505 | 356 | 354 | 0.16 |
| 3 | 26,395 | 19,889 | 3690 | 2345 | 769 | 359 | 217 | 0.19 |
| 4 | 31,371 | 25,147 | 4201 | 2957 | 415 | 497 | 332 | 0.17 |
| 5 | 35,274 | 28,075 | 6337 | 4428 | 1054 | 471 | 384 | 0.23 |
| 6 | 52,597 | 40,483 | 6416 | 2747 | 2279 | 650 | 740 | 0.16 |
| 7 | 60,525 | 48,719 | 11,484 | 5608 | 4038 | 1084 | 754 | 0.24 |

*Proportion assigned is based on number of localized signal.

## Sex triggered vocalization averages

The temporal relationship between male and female vocalizations was examined by calculating the average male vocalization rate centered on the time of each female vocalization. For every female vocalization, all male vocalizations within 30 s were partitioned into 1-s bins (60 bins total). Each bin was summed and normalized by the number of female vocalizations. The combined response of each recording session was then averaged. Note that sex triggered vocalization averages cover a 60 s window around the trigger vocalizations. We excluded potential trigger vocalizations that occurred within 30 s from the beginning or end of the data record (141 male and 3 female vocalizations).

To determine whether the male vocal rate was significantly higher than expected by chance, a comparison data set was generated for every recording session using a random permutation test. The times of the female vocalizations were randomly shuffled and used to calculate a shuffle-dependent male vocalization rate. This procedure removes any temporal correlation between the two sets of vocalizations while preserving both the temporal structure of the male vocalizations as well as the total number of vocalizations. The shuffle-dependent male vocalization rate for each group was then averaged to produce a mean shuffle-dependent vocalization rate. The random permutation process was repeated 1000 times to generate a distribution of average shuffle-dependent vocalization rates, which were used to determine the probability that the male vocalization rate was higher than expected by chance. For each of the 60 bins, the number of times that the actual data were above the shuffled data was calculated. These proportions were used with a binomial parameter estimate to determine significance. The actual data were considered significant at a conservative alpha <0.006 if at most one of the average shuffle-dependent vocalization rates was greater than the average from the actual data.

To examine the average female vocalization rate, the previously described analyses were also performed using male vocalizations as the trigger. To determine whether the vocalization rate changed significantly over time, a 1-way ANOVA was performed. Tukey multiple comparison tests were used to determine when the vocalization rate was significantly different from the other means.

## Vocal sequences

We called a sequence of vocalizations from a single male-female pair a vocal sequence if the male and female vocalized within 1 s of each other. The male and female mice participating in the vocal sequence were the vocal pair, while the other two mice were considered the non-vocal pair. This analysis was repeated for female-male vocal sequences. The behavior of the vocalizing and non-vocalizing mice was then examined separately for both male- and female-vocalization initiated events. We excluded vocal sequences that occurred within 30 s of the beginning or end of the data record, which removed 40 vocal sequences initiated by male vocalizations and 17 vocal sequences initiated by female vocalizations from the analysis. All vocal sequences were used when calculating the correlation between number of vocal sequences and number or length of chases (see details below).

## Distance between mice

For a 60 s window surrounding the onset of each vocal sequence, the position of the four mice was extracted from the tracked video files. At each frame, the Euclidean distance was calculated between

the vocally active mice. This procedure was repeated for the non-vocal mice in the same set of video frames. After determining the distances between the pair of vocal mice as well as the non-vocal mice for each vocal sequence, we calculated the average inter-mouse-distance for every recording session. The cumulative average for both the vocal and non-vocal mice was determined from the average for each of the seven groups. To determine whether the distance between the vocal pair of mice or the non-vocal pair of mice changed significantly over time, a 1-way ANOVA was performed. Tukey multiple comparison tests were used to determine when the distance was significantly different from the other means.

## Relative position of mice

The relative position of the mice during the vocal sequence was examined twenty-five seconds before and after and at the start of each sequence (pre, post and during). For each time point, the position and orientation of either the male or female mouse were translated and rotated such that the midpoint of the mouse's body was centered and the mouse was facing upwards. The position and orientation of the other mouse in the pair was translated and rotated by the same amount as its vocal partner. A 2D histogram was generated with the distribution of the relative positions of the vocal partner. The bin size for the x- and y-axes was 0.01 cm. A region of interest surrounding the midpoint of the centered mouse (circle with radius of 10 cm) was used to determine whether the distribution of relative positions differed between the anterior and posterior regions at each time point. For both the male- and female-centered relative positions, a chi-square analysis was performed on the proportion of anterior counts to total counts at the pre, during, and post time periods. Post-hoc Tukey-type multiple comparison tests were used to compare the three proportions for the male- or female-centered data.

## Speed of mice

For a 60-s window surrounding the onset of each vocal sequence, the instantaneous speed of the four mice was calculated between subsequent frames. The speed of each of the vocalizing mice was averaged to determine the mean instantaneous speed, which was then averaged across a recording session for all vocal sequences. This was repeated for non-vocal animals. The group average for both the vocal and non-vocal mice was determined from the average for each of the seven recording sessions. To determine whether the speed of the vocal pair of mice or the non-vocal pair of mice changed significantly over time, a 1-way ANOVA was performed. Tukey multiple comparison tests were used to determine when the speed was significantly different form the other means.

## Automated classifier

An automated classifier (Janelia Automatic Animal Behavior Annotator; JAABA; *Kabra et al. (2013)*, http://jaaba.sourceforge.net) was used for identifying events when a female mouse was being followed. The classifier was trained on 5228 frames of video (2341 positive examples and 2887 negative examples), using the mouse position data extracted by Motr. A subset of non-training frames (4040) was manually scored and compared to the output of the classifier to determine the accuracy of the system. The chase classifier had a false negative rate of 2.7% and a false positive rate of 7.5%. To determine which of the two males was following the identified female, we used a set of heuristics. First, the trajectories of the potential male participant had to overlap with that of the chased female by at least 20%. Second, the distance between the two mice at the start of the chase had to be within 20 cm. Consecutive chases involving the same participants were merged when the two events were separated by ~0.3 s or less. All chases needed to exceed ~0.17 s.

## Vocal interactions associated with chases

Vocal interactions that occurred during a chase were assigned to that chase if the vocalizing animals were participating in the chase. If the vocal interactions occurred within 30 s after the end of the previous chase, it was assigned to the previous chase. If the vocal interactions occurred within 30 s from the onset of the chase following the vocal interactions, then it was assigned to the following chase. If the vocal interactions were not within 30 s before or after the flanking chases, then it was not assigned to a chase. Vocal interaction rate during a chase was determined by dividing the number of vocal interactions in a chase by the duration of the chase. Vocal interaction rate outside a chase was determined by dividing the number

vocal interactions outside a chase by the difference between the time of the chase and the one-minute window surrounding the onset of the chase. To determine whether the vocal interaction rate during a chase was significantly different from the rate outside the chase, a Mann–Whitney U-test was performed. Central tendency and variability were reported with the median and interquartile range, respectively. The interquartile range was shown as the 75th–25th percentiles of the data.

### Female receptivity

Male vocalizations not followed by other vocalizations within 1 s were extracted and assigned to a chase if the vocalizing male was participating in the chase. These were classified as chases without vocal interactions. During chases, there were 107 male vocalizations that were not followed within a second by a female or unassigned vocalization and 823 vocal interactions. The average speed of the two vocally interacting animals at the time of the male vocalization was calculated as well as the speed of both the male and female. To determine whether the speed of the animals at the time of a male vocalization was significantly different between the two conditions, a Mann–Whitney U-test was performed.

## Acknowledgements

We thank K Seagraves, M Smear, E Pastalkova, K Branson, V Jayaraman, K Svoboda, S Sternson, J Simpson, A Leonardo, and R Neunuebel for comments on the manuscript, Vivarium and Advanced Computational and Technology staffs, and W Rowell for assistance.

## Additional information

### Funding

| Funder | Grant reference | Author |
| --- | --- | --- |
| Howard Hughes Medical Institute (HHMI) | Janelia Research Campus | Joshua P Neunuebel, Adam L Taylor, Ben J Arthur, SE Roian Egnor |

The funder had no role in study design, data collection and interpretation, or the decision to submit the work for publication.

### Author contributions

JPN, Conception and design, Acquisition of data, Analysis and interpretation of data, Drafting or revising the article; ALT, SERE, Conception and design, Analysis and interpretation of data, Drafting or revising the article; BJA, Wrote vocalization segmentation software, Analysis and interpretation of data, Drafting or revising the article

### Ethics

Animal experimentation: These experiments were performed at Janelia Research Campus in strict accordance with the recommendations in the Guide for the Care and Use of Laboratory Animals of the National Institutes of Health. All procedures were carried out according to a protocol approved by the Janelia Research Campus Institutional Animal Care and Use Committee (IACUC; protocol #11-70). The Janelia Research Campus Vivarium maintains full AAALAC accreditation.

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
