## [Decision Letter]

Thank you for sending your work entitled “Female mice ultrasonically interact with males during courtship displays” for consideration at *eLife*. Your article has been favorably evaluated by a Senior editor and three reviewers, one of whom, Peggy Mason, is a member of our Board of Reviewing Editors.

The Reviewing editor and the other reviewers discussed their comments before we reached this decision, and the Reviewing editor has assembled the following comments to help you prepare a revised submission.

The manuscript describes a valuable new method for localizing USVs in complex rodent groups of 4. This could prove to be a very useful method. The data supporting the accuracy of the method are largely impressive although some points could be clarified and others addressed.

There is importance as well in the demonstration that females vocally participate in mating interactions and that the females' USVs influence the course of the courtship. These are interesting and novel data that could open up new areas of inquiry.

In sum, this is a creative piece of work that looks at a hitherto ignored problem and has great promise for moving the field forward.

Some clarification of the rationale for particular analyses, and clearer descriptions of what some approaches entail, would make the manuscript more accessible for a general reader. Specific points that we ask the authors to clarify and/or address include:

The assigned calls are roughly ∼15% of the total number of calls that the mice made. The possibility of (unintentional) selection bias should be addressed. The authors are asked to illuminate the major reasons for exclusion. Maybe the authors can clarify if all calls between two time points were analyzed, and if only calls recorded before and after these two time points were rejected. It might also be possible to overlay “unambiguously assigned” call and "rejected" calls for a single recording session in order to demonstrate how assignments were distributed.

Explain the rationale for single housing animals. Were females single housed? Furthermore, the description of mice as housed in social groups is a little confusing, since they are housed individually until the briefer 5-hour social experiment; please clarify this in the Results section. It is an important clarification, since individual housing followed by social interaction drastically increases vocalization rate in mice in some paradigms. Related to this, did calls vary in rate from the start to the end of the 5-hour period?

Comparisons should be made between male rate wi/1s of male vocalizing vs wi/1s of female vocalizing and similar comparison for female. Without such comparisons, the statement “mice increased their vocalization rate whenever the opposite sex vocalized” does not appear warranted. Comparing with 1s of another vocalization with nothing happening is just not the right comparison.

Additional information about the chases could enrich the results and conclusions e.g. incidence of mounting or lordosis, consistency of the specific individuals involved.

In the eighth paragraph of the subsection headed “Vocalizations during complex social interactions”, the authors use a statistical argument on the error rate to support the argument that they are correctly attributing calls to females. The first set of data they use is understandable: they use all interactions, and estimate how many vocalizations could be incorrectly assigned to females based on the error rate. In the second part of the paragraph, they focus on vocal interactions occurring during chases. Here, they state that they assign 'interactions' incorrectly based on error rate. How is it possible to assign an exchange, as opposed to a vocalization, to a particular individual female?

Minor comments:

Reviewer #1

How can video tracking at 29 Hz be used to accurately synch to audio information analyzed in 5 ms bins? How was location estimated between snapshots taken at 29 Hz and then distributed to 200Hz sound source estimates?

Reviewer #2

1) Figure 3 seems to suggest that females produce only a fraction of what males utter which seems to put the initial statement into perspective. Please provide an explanation of this seemingly contradicting result.

2) Figure 5 appears most relevant in supporting the claim that male-female vocal interaction help coordinate movements between the two animals. I do not understand, however, why non-vocal pairs show also a small peak around zero time (Figure 5). Should it not be a flat curve, just like the distance curve between mates (Figure 5)?

3) Figure 6 again, appears to counter the previous observation that male-female vocal interactions are used to coordinate movements because differences between vocal and non-vocal interactions are small (Figure 6). Maybe it is possible to demonstrate the difference with an example of a representative recording.

4) It would be helpful to compare male and female vocalizations for spectro-temporal sex differences and revisit the claim made in previous studies that there are no differences in acoustic features. The lack of differences in previous studies may have been the result of different (less natural?) experimental setup.

Reviewer #3

1) Although the emphasis of the manuscript is on ultrasonic vocalizations, it would be helpful to have some statement near the start distinguishing female ultrasonic and audible vocalizations, particularly since females produce audible vocalizations during interactions with males.

2) In addition to the important error estimates that the authors make to support the attribution of calls to females, is any other type of corroborating evidence possible? For example, are there any instances of overlap in ultrasonic calls for 2 mice in close proximity? This would be a pretty powerful, albeit anecdotal, class of evidence. If no overlap is ever observed, does that mean that calls are alternating? This would also be quite interesting.

---

## [Author Response]

*The assigned calls are roughly ∼15% of the total number of calls that the mice made. The possibility of (unintentional) selection bias should be addressed. The authors are asked to illuminate the major reasons for exclusion. Maybe the authors can clarify if all calls between two time points were analyzed, and if only calls recorded before and after these two time points were rejected. It might also be possible to overlay “unambiguously assigned” call and "rejected" calls for a single recording session in order to demonstrate how assignments were distributed*.

We have created a new figure, Figure 4, which addresses the ways in which assignment might be biased: in time (Figure 4), space (Figure 4) and relative mouse position (Figure 4). Figure 4 plots assigned (red bars) and unassigned (gray bars) vocalizations as a function of time, showing that in all groups vocalizations are assigned throughout the experiment, and there is therefore no temporal bias. Figure 4 shows that vocalizations (both assigned and unassigned) are less likely to be detected in the corners of the box (where the signal-to-noise ratio may not be as favorable on the two near wall microphones), illustrating the need for additional microphones to cover the space completely, but also showing that there is no spatial bias in assignment. Figure 4 shows the relative distances between the closest mouse to the sound source (x-axis) and the second closest mouse (y-axis), plotted as the difference between the distance distribution (red = higher proportion of assigned vocalizations; blue = higher proportion of unassigned vocalizations. Assigned vocalizations occurred predominantly when the estimated sound source is close to one mouse, whereas unassigned vocalizations occur when two mice are equidistant from the estimated sound source. This suggests that ambiguity in identifying the source of the signal is the major reason for excluding signals.

We have added a paragraph to the manuscript describing this figure, and also a figure legend. We have also changed the subsequent figure numbers in the text, figures, and legends.

In the subsection headed “Vocalizations during complex social interactions”, we state:

“A possibility exists that the sound source localization system introduces a selection bias because the majority of vocalizations are unassigned. […] With this system we were able to follow the vocal and physical interactions of mice in large, mixed sex groups and precisely distinguish sex-specific contributions during social interactions.”

In addition, we have changed and rephrased the legend for Figure 4.

*Explain the rationale for single housing animals. Were females single housed? Furthermore, the description of mice as housed in social groups is a little confusing, since they are housed individually until the briefer 5-hour social experiment; please clarify this in the Results section*. *It is an important clarification, since individual housing followed by social interaction drastically increases vocalization rate in mice in some paradigms. Related to this, did calls vary in rate from the start to the end of the 5-hour period?*

The reviewers bring up an excellent point. Housing type has an enormous effect on mouse social behavior. We have added the following text to emphasize this (in the “Vocalizations during complex social interactions” subsection):

“Because group housing affects social behavior in males (38) and females (44), all mice were singly housed for at least 14 days before the experiment in an attempt to maximize the amount of social information exchanged after the animals were introduced to each other.”

To address the separate question of variation in call rate over the course of the experiment, we have added Figure 4 showing the total number of vocalizations detected in all 7 groups of animals over the course of the experiment.

*Comparisons should be made between male rate wi/1s of male vocalizing vs wi/1s of female vocalizing and similar comparison for female. Without such comparisons, the statement “mice increased their vocalization rate whenever the opposite sex vocalized” does not appear warranted. Comparing with 1s of another vocalization with nothing happening is just not the right comparison*.

There are two types of analyses that address the temporal relationship between male and female vocalizations. The first (illustrated in Figure 4 in the original manuscript, now Figure 5 in the revised manuscript) looks at all vocalizations of the opposite sex within a +/- 30s temporal window centered on each vocalization. This analysis shows a clear peak close to zero, indicating that when a male vocalization occurred, the probability of a female vocalization peaked, and similarly when a female vocalization occurred, the probability of a male vocalization peaked. This is the basis of our statement that mice increased their vocalization rate whenever the opposite sex vocalized. The control for this analysis was not nothing happening, but rather a random shuffling of vocalization times (which produces no peak). The second type of analysis was based on the observation of this peak. This analysis involved specific comparisons of male and female vocalizations within 1s (Figures 5 and 6).

Because the focus of this study is on male-female interactions during courtship, we focused on the temporal relationship between male and female vocalizations rather than male-male or female-female (which we plan to address in future studies). However, to clarify the results of our analysis we have reworded the text.

In the Introduction, by removing specific details:

“Furthermore, we discovered a temporal correlation between male and female vocalizations.”

In the Results, by modifying the text to read:

“…when male vocalizations occurred, the probability of female vocalizations peaked. Similarly when female vocalizations occurred, the probability of male vocalizations peaked.”

*Additional information about the chases could enrich the results and conclusions e.g. incidence of mounting or lordosis, consistency of the specific individuals involved*.

While we did score mounting events, they only occurred in two of the seven cages, so we didn’t feel that we had a sufficient sample size to address this issue in detail. For future experiments we will be allowing the cages to run for much longer—until the birth of pups—, and we feel such experiments, which include the tangible result of the courtship process, will be a more suitable forum to address male-female chases in detail. However, to provide readers with more information about chases, we have modified original Figure 6 (current Figure 7) to indicate the identity of the male-female pair (e.g. male 1 and female 2 from group 4) and have rephrased the legend for Figure 7 and added the following to the manuscript (subsection headed “Vocalizations during complex social interactions”):

“The proportion of chases performed by each male varied across groups. […] the number of male-initiated vocal interactions was strongly correlated with both total chase time (Figure 7, r=0.74, p<10^-5^) and number of chases (r=0.55, p<0.003).”

In the eighth paragraph of the subsection headed “Vocalizations during complex social interactions”, the authors use a statistical argument on the error rate to support the argument that they are correctly attributing calls to females. The first set of data they use is understandable: they use all interactions, and estimate how many vocalizations could be incorrectly assigned to females based on the error rate. In the second part of the paragraph, they focus on vocal interactions occurring during chases. Here, they state that they assign 'interactions' incorrectly based on error rate. How is it possible to assign an exchange, as opposed to a vocalization, to a particular individual female?

The reviewers are correct. We mis-calculated the expected number of chase-associated errors (on a per-interaction rather than per-vocalization basis). If we recalculate on a per-vocalization basis then the 823 vocal interactions that occurred during chases would yield 823 assigned male vocalizations and 823 assigned female vocalizations for a total of 1646 vocalizations. When calculating the number of vocal interactions that were incorrectly identified, we would only expect 115 false positives (not 58, as stated in the original manuscript). 115 false positives still leaves 708 correctly identified male and female vocal exchanges. However, we feel the reviewers are correct, that calculating an error rate in this way is potentially inappropriate. We have therefore removed this portion of the argument, leaving only the total per-vocalization argument.

*Minor comments*:

Reviewer #1

How can video tracking at 29 Hz be used to accurately synch to audio information analyzed in 5ms bins? How was location estimated between snapshots taken at 29 Hz and then distributed to 200Hz sound source estimates?

The reviewer is correct to be confused by this. Because video data was collected at 29 Hz, each frame is roughly 34.48ms long, so each frame is roughly 7 snippets. The position of the mice is taken per frame, so all snippets within a frame have the same position. When mice are moving quickly, this will have the effect of introducing a difference between the video estimate of mouse position and the snippet-based estimate. However, we have modified to methods to emphasize that the frequency of localization is 29 Hz, not 200 Hz

Reviewer #2

*1)*
Figure 3
*seems to suggest that females produce only a fraction of what males utter which seems to put the initial statement into perspective. Please provide an explanation of this seemingly contradicting result*.

The reviewer very correctly points out that we show that males produce many more vocalizations in a heterosexual context than females, with females producing only approximately 25% of the number of vocalizations produced by males. There are two reasons we believe that this is not a contradictory result and we have attempted to emphasize these two points more clearly in the text: First, because previous researchers have shown that female mice can produce high rates of vocalizations when paired with other females (Sales, 1972; [47]), and because the acoustic structure of female vocalizations has been shown to be similar to that of males (Hammerschmidt et al., 2012), it has not previously been possible to determine what fraction (if any) of the vocalizations emitted during male-female interactions were produced by the female. And, secondly, rather than female vocalizations being produced independently of male vocalizations (albeit at a lower rate), female vocalization timing is coordinated with that of males (as shown in Figures 5 and 6).

*2)*
Figure 5
*appears most relevant in supporting the claim that male-female vocal interaction help coordinate movements between the two animals. I do not understand, however, why non-vocal pairs show also a small peak around zero time (*Figure 5*). Should it not be a flat curve, just like the distance curve between mates (*Figure 5*)?*

We believe, since all four animals are in the same space, that the high speed of the vocally interacting pair produces a slight increase in the speed of the non-vocally interacting pair, since the chase has some probability of going near the non-interacting pair. This slight peak, however, is not significant. We have added the following sentence to the Results section to address the reviewer’s observation:

“Despite a slight peak, the speed of the non-vocal pair did not change significantly over time (one-way ANOVA, F1740,10446=1.0, p>0.9). We believe this peak occurs because all four mice are in the same cage, and therefore the behavior of the vocal pair may affect the behavior of the non-vocal pair.”

*3)*
Figure 6
*again, appears to counter the previous observation that male-female vocal interactions are used to coordinate movements because differences between vocal and non-vocal interactions are small (*Figure 6*). Maybe it is possible to demonstrate the difference with an example of a representative recording*.

It is true that there is little difference (in fact, no significant difference, p>0.25) in male speed between chases in which a male-female vocal interaction occurred (VI) and chases where only the male vocalized (NVI). However, female speed is quite different depending on whether or not a vocal interaction occured (p<p^-10^), such that the speed at the 50th percentile in the VI case is almost double that in the NVI case. We believe that plotting the average speed in the original figure (original Figure 6) led to misunderstanding, as was explicitly pointed out in the main comments, and we have therefore removed it.

*4) It would be helpful to compare male and female vocalizations for spectro-temporal sex differences and revisit the claim made in previous studies that there are no differences in acoustic features. The lack of differences in previous studies may have been the result of different (less natural?) experimental setup*.

We agree with the reviewer that this is an interesting area for future experiments, but feel that the necessary analyses are too involved for inclusion in this manuscript.

Reviewer #3

*1) Although the emphasis of the manuscript is on ultrasonic vocalizations, it would be helpful to have some statement near the start distinguishing female ultrasonic and audible vocalizations, particularly since females produce audible vocalizations during interactions with males*.

We believe it is clear, from the Title, Abstract and Introduction, that we are only addressing ultrasonic vocalizations. Both male and female mice also produce audible vocalizations during stressful interactions (fights, unwanted courtship attempts, restraint by humans), but we are not addressing them in this manuscript.

*2) In addition to the important error estimates that the authors make to support the attribution of calls to females, is any other type of corroborating evidence possible? For example, are there any instances of overlap in ultrasonic calls for 2 mice in close proximity? This would be a pretty powerful, albeit anecdotal, class of evidence. If no overlap is ever observed, does that mean that calls are alternating? This would also be quite interesting*.

We did observe some cases of overlapping vocalizations. The majority of the time these overlapping vocalizations were unlocalizable (either because the mice were close and equidistant, or the overlapped vocalization did not form a tight enough cluster), and therefore were excluded from analysis. We have added a statement to that effect to the Methods section. We definitely agree that the exact temporal relationship between calls is a fascinating one, which we plan to address in a future manuscript, but we believe that the system needs additional improvements in assignment percentage before a more precise temporal analysis is appropriate. We plan to improve the system in future by doubling the number of microphones (from 4 to 8) and experimenting with different arena shapes (notably, a circle, which should lack the ‘dead’ spots in the current square arena). In addition, we plan to explore the suggestion of Reviewer 1, of adapting snippet frequency bandwidth to the local structure of the vocalization to see if that can also improve accuracy and resolution.